# Creation of a Prognostic Model Using Cuproptosis-Associated Long Noncoding RNAs in Hepatocellular Carcinoma

**DOI:** 10.3390/ijms24129987

**Published:** 2023-06-10

**Authors:** Lihong Yang, Xiao Jia, Yueyue Fu, Jiao Tian, Yijin Liu, Jianping Lin

**Affiliations:** State Key Laboratory of Medicinal Chemical Biology, College of Pharmacy and Tianjin Key Laboratory of Molecular Drug Research, Nankai University, Tianjin 300000, China; 1120200609@mail.nankai.edu.cn (L.Y.); 1120210643@mail.nankai.edu.cn (X.J.); 2120211398@mail.nankai.edu.cn (Y.F.); 1120210647@mail.nankai.edu.cn (J.T.)

**Keywords:** hepatocellular carcinoma, cuproptosis, LncRNA, risk model, PCAT6

## Abstract

Cuproptosis is an unusual form of cell death caused by copper accumulation in mitochondria. Cuproptosis is associated with hepatocellular carcinoma (HCC). Long noncoding RNAs (LncRNAs) have been shown to be effective prognostic biomarkers, yet the link between lncRNAs and cuproptosis remains unclear. We aimed to build a prognostic model of lncRNA risk and explore potential biomarkers of cuproptosis in HCC. Pearson correlations were used to derive lncRNAs co-expressed in cuproptosis. The model was constructed using Cox, Lasso, and multivariate Cox regressions. Kaplan–Meier survival analysis, principal components analysis, receiver operating characteristic curve, and nomogram analyses were carried out for validation. Seven lncRNAs were identified as prognostic factors. A risk model was an independent prognostic predictor. Among these seven lncRNAs, prostate cancer associated transcript 6 (PCAT6) is highly expressed in different types of cancer, activating Wnt, PI3K/Akt/mTOR, and other pathways; therefore, we performed further functional validation of PCAT6 in HCC. Reverse transcription–polymerase chain reaction results showed that PCAT6 was aberrantly highly expressed in HCC cell lines (HepG2 and Hep3B) compared to LO2 (normal hepatocytes). When its expression was knocked down, cells proliferated and migrated less. PCAT6 might be a potential biomarker for predicting prognosis in HCC.

## 1. Introduction

Liver cancer is one of the deadliest cancers, the death rate of which has been constantly increasing for decades [1]. Despite the advances in treatment, treating liver cancer is still challenging [2,3]. For early stage patients with liver cancer, surgery and liver transplantation are effective treatments; however, the recurrence rate is more than 70% after five years [4,5]. The efficacy of chemotherapy and radiotherapy is poor for patients in the advanced stage of liver cancer; thus, their survival rate is relatively low [5,6]; therefore, the discovery of more diagnostic and prognostic biomarkers is urgent to improve the long-term prognosis of liver cancer patients.

Copper is an indispensable cofactor in several biological processes, maintaining a steady state of low concentrations in the body [7,8]; however, abnormal copper homeostasis leads to the occurrence of tumors and other various diseases [7,9,10]. Several studies concluded that the content of copper in malignant tumors is significantly higher than that in normal tissues [11]. Moreover, a moderate increase in intracellular copper concentration can cause cytotoxicity and even cell death [12]. Copper-dependent regulation of cell death mainly relies on mitochondrial respiration. Through direct binding of copper to the thiolated components of the tricarboxylic acid cycle, it induces the aggregation of acylated proteins and the loss of iron sulfur cluster proteins, leading to protein toxicity stress and ultimately cell death. The copper ionophores induced cell death [13,14,15] and a process named cuproptosis. Cuproptosis is a regulated form of cell death triggered by an excess of copper. It is different from conventional ferroptosis, apoptosis, pyroptosis, and necrosis, in that intracellular copper binds to lipases in the tricarboxylic acid cycle, which in turn leads to the aggregation of copper-bound lipidated mitochondrial proteins and a reduction in Fe-S (iron-sulfur) clusters, which in turn leads to proteotoxic stress and cell death [15]. It has recently been suggested that elevated redox-active free copper ions may be associated with hepatocellular carcinoma (HCC) [16]. Copper oxide nanoparticles may induce apoptosis by activating p53 and caspase-3 in HCC cells, thereby reducing the potential of the mitochondrial membrane [17]. Thus, cuproptosis may be closely associated with HCC, and cuproptosis may be a potential therapeutic modality for HCC. The existing studies mainly focused on the role of protein-coding genes in the occurrence of cuproptosis-related genes, while the research on the regulation mechanism of non-coding RNAs on cuproptosis-related genes is relatively limited [15,18,19,20].

About 90% of human genomic DNA has been transcribed, of which only 2% encodes proteins, and the rest are non-coding RNAs (ncRNAs). Among them, 70% of ncRNAs are long non-coding RNAs (lncRNAs), which are more than 200 nucleotides in length and have no protein-coding capacity [21,22]. The lncRNAs related to liver cancer regulate the transcriptional, epigenetic, and post-transcriptional levels of genes by binding with DNA, protein, and RNA [23,24]. Additionally, aberrantly expressed lncRNAs play a key role in the occurrence and development of continuous tumor proliferation, invasion, and angiogenesis [23,25]. Several studies have confirmed that lncRNAs are aberrantly expressed in HCC and may play a key role in the development of HCC. For example, highly upregulated in liver cancer (HULC) is located at is located on the chromosome 6p24.3, producing a long RNA that was discovered as upregulated in HCC and is associated with cancer progression [26]. WRAP53 in peripheral blood is an independent prognostic marker predicting high recurrence rates in HCC patients [23]. In addition, studies have also confirmed that lncRNAs may be a prognostic factor in cuproptosis in a variety of cancers. For example, a risk model constructed based on eight lncRNAs, including RNF139-AS1, LINC00996, NR2F2-AS1, AL590428.1, SEC24B-AS1, AC006566.1, UBE2Q1-AS1, and AL021978.1, could be used as an independent prognostic indicator for bladder urothelial carcinoma [27]. The same lncRNA profile associated with cuproptosis could be used to predict the prognosis and immune relevance of kidney renal papillary cell carcinoma, providing new potential therapeutic targets and prognostic markers for kidney renal papillary cell carcinoma patients [28]; however, the regulation mechanism of lncRNAs on cuproptosis-related genes in HCC is still unclear. Further analysis of cuprotosis-related lncRNAs can provide more diagnostic, prognostic, and therapeutic targets for patients with liver cancer; therefore, the identification of lncRNAs related to cuproptosis-related genes is of great significance for the clarification of underlying mechanisms of hepatocarcinogenesis and the investigation of new therapeutic targets.

In this study, we aimed to construct a prognostic risk model based on lncRNAs associated with cuproptosis in HCC. We found that of the seven lncRNAs modeled, prostate cancer associated transcript 6 (PCAT6) is highly expressed in various cancers and may activate Wnt, PI3K/Akt/mTOR, and other pathways to regulate cell proliferation and migration; therefore, we hope to further explore the prognostic value of PCAT6 through experiments. The results of this study may help to improve the individualized treatment and prognostic assessment of patients with liver cancer.

## 2. Results

### 2.1. Construction and Validation of the Risk Model

Co-expressed lncRNAs associated with cuproptosis were obtained by Pearson correlation analysis. The parameters were set to |pearson| > 0.3 and *p*_value_ < 0.001. Through the Pearson correlation analysis, 2444 lncRNAs related to cuproptosis-related genes were obtained, and the correlation between cuproptosis-related genes and lncRNAs was visualized by the Sankey diagram (Figure 1A). Then, 31 lncRNAs related to patient prognosis were obtained by univariate Cox regression analysis (Figure 1B). Eleven lncRNAs were obtained by lasso regression analysis [29], which can be used to identify the most effective predictive markers and generate prognostic indicators for the prediction of clinical outcomes (Figure 1C,D). Seven cuproptosis-related lncRNAs were determined as independent prognostic factors by multivariate Cox regression identified (Figure 1E), which was used to construct the risk model.

The training set (Figure 2A–C), testing set (Figure 2D–F), and entire set (Figure 2G–I) were used to evaluate the grouping ability of the risk model. Table 1 shows that the dataset was feasibly independent of clinical characteristics. As shown in the risk score map, in the training set, testing set and entire data set of the cancer genome atlas (TCGA), the death rate of patients in the high-risk group was significantly higher than that in the low-risk group with increasing risk scores (Figure 2A,B,D,E,G,H). As shown in Figure 2C,F,I, in the training set, testing set, and entire data set of TCGA, the Kaplan–Meier (K-M) curves were used to verify the prognostic ability of the risk model, and the survival rate of the high-risk group was significantly lower than that of the low-risk group. Meanwhile, the progression-free survival (PFS), disease-specific survival (DSS), and disease-free survival (DFS) of the HCC patients were verified as outlined below. As shown in Figure 2J–L, the PFS, DFS and DSS of patients in the high-risk group were significantly lower than those in the low-risk group. Principal components analysis (PCA) was used to further verify the grouping ability of the risk model. The high-risk and low-risk groups obtained by the entire gene expression profiles, cuproptosis-related genes, and lncRNAs of cuproptosis-related genes were difficult to distinguish (Appendix A), while the high-risk and low-risk groups divided by the risk model can be clearly distinguished (Appendix A).

### 2.2. Independent Prognostic Validation of the Risk Model

Univariate and multivariate Cox regression analyses were used to determine whether the risk models were independent prognostic models. The hazard ratio (HR) and 95% confidence interval (CI) of the risk model through univariate Cox regression analysis was 1.068 (1.039–1.098, *p*_value_ < 0.001), and the HR and 95% CI of the risk model through multivariate Cox regression analysis were 1.063 (1.030–1.096, *p*_value_ < 0.001), which indicates that the risk model can be used as an independent prognostic marker (Figure 3A,B). In addition, the receiver operating characteristic curve (ROC) and concordance index (C-index) reflected that the risk model had better sensitivity and prediction accuracy than clinical characteristics (age, gender, stage, grade, T stage) (Figure 3C–E).

### 2.3. Association between Risk Model and Clinical Characteristics

Next, we assessed the correlation between the risk model and clinical characteristics. As shown in Figure 4A, the risk model was significantly correlated with the grade, stage and T stage of the HCC, but not with other clinical features. As shown in Figure 4B–D, grades 2–3, stages II–III, and T2–T3 of patients in the high-risk group were significantly higher than those in the low-risk group. At the same time, we analyzed the clinical prognosis between high-risk groups and low-risk groups. As shown in Figure 5A–J, the survival rate for clinical characteristics (age, stage, grade, gender and T stage) of patients in the high-risk group was significantly lower than that in the low-risk group.

### 2.4. Immune Infiltration and Immunotherapy Response Analysis of Risk Model

Traditional gene expression analysis strategies focus on identifying individual genes that exhibit differences between two states; however, they cannot detect biological processes. Therefore, gene set enrichment analysis (GSEA) [30] was performed on the high-risk and low-risk groups to further validate the enrichment of relevant signaling pathways in the risk model. Pathways with NES > 1 and *p*_value_ < 0.05 may be statistically significant signal pathways. As shown in Appendix A, the high-risk group was enriched with pathways in cancer, cell cycle, adherens junction, MAPK signaling pathway, VEGF signaling pathway, and Wnt signaling pathway, indicating that the tumors of patients in the high-risk group were more proliferative and aggressive. Appendix A indicated that the high-risk group was involved in immune- and inflammation-related signaling pathways (Fc gamma R-mediated phagocytosis, toll-like receptor signaling pathway, B cell receptor signaling pathway, NOD-like receptor signaling pathway, natural killer cell-mediated cytotoxicity, complement, and coagulation cascades). In conclusion, the above results indicated that patients in the high-risk group were closely related to the occurrence, development, and immunity of HCC.

To further investigate the correlation between the risk model and immune infiltration, TIMER2.0 and TISIDB data were used to analyze whether tumor-infiltrating immunocytes (TIICs) and tumor-infiltrating lymphocytes (TILs) were different between high-risk and low-risk groups. As shown in Figure 6A, immune cell infiltration was different between high-risk and low-risk groups. Based on the genetic expression spectrum of tumor samples, combined with the marker genes of various immune cells confirmed by studies, it could be observed using the ssGSEA algorithm that some immune cells had differences under different samples. Studies have shown that tumor infiltration of B cells and pulp cells have a key synergistic role in inhibiting the development of tumors. The synergy promotes the antitumor immunity of T cells through its unique antigen to show better prognosis for patients [31]. Figure 6B shows that the activated CD8 T cells and activated B cells of the low-risk group were higher than that of high-risk group, explaining why the prognosis of the low-risk group patients was better than that of the high-risk group. The ESTIMATE algorithm was used to analyze stromal scores and immune scores for patients in high-risk and low-risk groups. The tumor microenvironment (TME) includes tumor cells, immune cells, cancer-associated fibroblasts (CAF), endothelial cells (EC), and other cell types. They play an important role in the development of tumors. Studies have shown that the higher stromal scores and immune scores of HCC patients are related to a good prognosis [31,32]. As shown in Figure 6C, the TME scores of the high-risk groups are lower than that of the low-risk groups, which means that the tumor purity was very high, the content of immune cells and matrix cells was low, and the prognosis of patients was poor.

Next, the immunotherapy response of the risk model was analyzed by evaluating immune checkpoints, tumor mutation burden (TMB), and tumor immune dysfunction and exclusion (TIDE). In the tumor microenvironment, the immune system can recognize tumor cells and play an important role in the occurrence and development of tumors. To date, immunotherapy has become a promising treatment method for liver cancer, and the blockade of immune checkpoints is one of the most promising methods in tumor immunotherapy. We preliminarily analyzed the differential expression of immune checkpoint genes in the risk model. As shown in Figure 6D, the expression level of immune checkpoint genes in the high-risk group was different from that in the low-risk group. Among them, the expression level of programmed death-ligand 1 (PD-L1, also called B7-H1 or CD274) and checkpoint cytotoxic T lymphocyte-associated antigen 4 (CTLA4) was higher in the high-risk group, which implied that the high-risk group may be more effective with anti-PD-L1 and anti-CTLA4 therapy. TIDE was used to assess the likelihood of tumor immune escape in tumor samples to predict the response to immune checkpoint blockade [33]. As shown in Appendix A, the TIDE scores of patients in the low-risk group were significantly higher than those in the high-risk group, suggesting that patients in the high-risk group could benefit more from immune checkpoint therapy. Studies have shown that TMB is a new biomarker for predicting the response to immune checkpoint therapy [34]. As shown in Appendix A, the TMB of patients in the high-risk group was significantly higher than those in the low-risk group, and the waterfall plot presented the 15 genes with the most mutation (Appendix A).

### 2.5. Sensitivity of Risk Models to Anticancer Drugs

To identify potential drugs for the treatment of HCC patients using the proposed risk model, the pRRophetic algorithm was used to predict the susceptibility of patients in the high-risk or low-risk groups to anticancer drugs. A total of 119 anticancer drugs (Appendix A) were obtained, and the patients in the high-risk group were predicted to be more sensitive to anticancer drugs. Figure 7A–H shows the anticancer drugs. To verify that the screened drug was effective against HCC, sorafenib was used for bioassay validation. The half maximal inhibitory concentration (IC_50_) of sorafenib in the HCC cell line (Huh7 cells) was 5.694 μM (Appendix A), and sorafenib was used at concentrations of 0 μM, 2.5 μM, 5 μM, and 10 μM for the corresponding experiments in the subsequent biologic validation. As shown in Appendix A, the proliferation of Huh7 cells received a significant inhibition with increasing sorafenib drug concentration and increasing treatment time. Carboxyfluorescein succinimidyl ester (CFSE) freely penetrates the cell membrane, and during cell division and proliferation, its fluorescence is equally distributed to the two daughter cells, resulting in a stepwise decrease in fluorescence intensity with cell division, and based on this property, it can be used to detect cell proliferation. As shown in Appendix A, the fluorescence intensity of the cells in the sorafenib-treated group was significantly higher than that of the control cells, and the inhibition of Huh7 cells increased with the increase in concentration. At the same time, cell cycle assays were used to verify whether sorafenib would produce a block in the cycle of Huh7 cells. As shown in Appendix A, the proportion of Huh7 cells in G1 phase increased and the proportion of S phase decreased after the addition of sorafenib. In conclusion, sorafenib, one of the predicted drugs, had some antiproliferative effect on HCC cell lines.

### 2.6. Construction and Assessment of Nomogram

We constructed a nomogram combining the risk model with clinical characteristics to assess 1-year, 3-year, and 5-year survival in patients with HCC (Figure 8A). The nomogram calibration plot showed that the array plot exhibited high predictive accuracy (Figure 8B), indicating that the accuracy of array plot predictions was good.

### 2.7. Potential lncRNA Biomarkers and Targets for HCC

We further analyzed the effects of seven lncRNAs alone on HCC. As shown in Figure 9 and Appendix A, the expression level of lncRNAs (TMCC1-AS1, PCAT6, ACVR2B-AS1) was significantly associated with the prognosis of HCC, and patients with high lncRNAs expression had poor survival rates (*p*_value_ < 0.001); however, the expression level of lncRNAs (AC092418.2, SACS-AS1) was not associated with the prognosis of HCC. Next, through further literature research, we found that research on TMCC1-AS1 and ACVR2B-AS1 related to cancer was rarely reported. PCAT6 was highly expressed in various cancers and may activate Wnt, PI3K/Akt/mTOR, and other pathways to regulate cell proliferation and migration; however, the mechanism of action of PCAT6 in HCC is unclear. Therefore, we performed further functional validation of PCAT6 in HCC.

### 2.8. Inhibition of PCAT6 Inhibits the Proliferation and Migratory Capacity of HCC Cells

To investigate whether PCAT6 is aberrantly expressed in HCC, we determined the levels of PCAT6 in normal liver cells (LO2 cells) and two HCC cell lines by RT-qPCR. The results suggested that PCAT6 expression was significantly upregulated in both HepG2 cells and Hep3B cells compared to LO2 (Appendix A). To further investigate the function of PCAT6 in HCC, we designed two siRNAs. RT-qPCR assay results showed that PCAT6 levels were most significantly reduced in HepG2 cells after the effect of transfection with si-PCAT6 (Appendix A). The results of the CCK8 experiments suggested that silencing PCAT6 expression significantly inhibited the proliferative capacity of HepG2 cells compared to the control (Appendix A). Cell scratch results consistently showed that cell migration was significantly inhibited upon transfection with si-PCAT6 compared to the control (Appendix A). All these results confirm that silencing PCAT6 inhibits the HCC cell proliferation and migratory capacity in vitro.

## 3. Discussion

A new form of copper-induced cell death, termed cuproptosis, has recently been proposed to occur in human cells. It has been proposed that copper oxide nanoparticles may induce apoptosis by activating p53 and caspase-3 in HCC cells, thereby reducing the potential of the mitochondrial membrane. Thus, cuproptosis may be closely associated with HCC, and it is of great interest to explore the lncRNAs associated with cuproptosis in HCC and the potential role of cuproptosis-related genes and lncRNAs in HCC and related mechanisms, which is beneficial to the exploration of novel therapeutic targets and drug development for HCC. The risk model was constructed based on seven cuproptosis-related lncRNAs, and the expression of the lncRNAs was higher in the high-risk group than in the low-risk group. Among them, TMCC1-AS1 is closely related to the proliferation and invasion of liver cancer [35] and may be a potential therapeutic target for liver cancer [35,36]. ACVR2B-AS1 is highly expressed and associated with poor prognosis in liver cancer [37]. In HCC, PCAT6 may inhibit cancer development by regulating the expression of hnRNPA2B1 through miR-326. PCAT6 may be closely associated with the development of HCC [38]. SACS-AS1 is correlated with the development of bladder cancer [39]. Little research has been reported on the association of another three lncRNAs (AC084083.1, AL021154.1, AC092418.2) with HCC. The results of the ROC curve and C-index analysis indicated that the accuracy of the risk model was better than that of the clinical characteristics. Our nomogram results are consistent with the ROC and C-index results. The 1-, 3-, and 5-year survival rates predicted by the nomogram were the same as the observed survival rates and could provide new insights into the prognosis, diagnosis, and treatment of liver cancer.

The high-risk group was enriched with more signaling pathways related to tumor proliferation and invasion by analyzing the GSEA of the risk model, suggesting that the high-risk patients had a poor prognosis amongst HCC patients. Moreover, the high-risk group was enriched with more immune-related signaling pathways, indicating that there may be differences in immune infiltration between high-risk and low-risk groups. The immune infiltration results based on TIMER, CIBERSORT, quantTIseq, xCell, MCP-counter, and EPIC algorithms showed that immune infiltration cells differed between high-risk and low-risk groups. Immune checkpoint blockade (ICB) for cancer treatment could lead to long-term clinical benefits [40]. Differences in immune infiltration between high-risk and low-risk groups suggested different degrees of benefit from immunotherapy. Previous studies suggested that TMB and TIDE can predict the treatment of patients with immune checkpoint inhibitors [33,34]. Moreover, the high-risk group had higher TMB and lower TIDE scores, indicating that patients in the high-risk group benefited more from immunotherapy.

PCAT6 is located on chromosome 1q32.1, alias KDM5B-AS1, KDM5BAS1, PCAN-R1, ncRNA-a2 or onco-lncRNA-96 [41]. Several studies have now confirmed that PCAT6 is not only highly expressed in a variety of human cancers but may also act through miRNA sponges to activate the Wnt/β-catenin signaling pathway and PI3K/Akt/mTOR pathway, which in turn affects cell proliferation, migration, invasion, cycle, and apoptosis [42,43]. For example, in non-small-cell lung cancer cells, knocking down the expression of PCAT6 significantly inhibited the proliferation and migration of lung cancer cells [44]. In Eca-109 and Kyse-30 cells, the knockdown of PCAT6 promoted apoptosis in esophageal squamous cell carcinoma [45]. Functional assays in Lang C’s studies confirmed that PCAT6 knockdown also significantly inhibited PCa cell invasion, migration, and proliferation in vitro, bone metastasis, and tumor growth in vivo [46]; however, the mechanism of action of PCAT6 in HCC is unclear. In the present study, we found that PCAT6 may be co-expressed with LIPT1, LIPT2, SLC31A1, GLS, etc., by co-expression analysis of copper death genes and lncRNA in HCC; however, how exactly the two interact with each other and how they affect copper death still needs further experimental investigation. In this study, bioinformatic tools, such as univariate and multifactorial COX regression and the construction of prognostic models, initially suggested that PCAT6 might be a potential prognostic predictor of HCC. In addition, we measured PCAT6 levels in normal hepatocytes and a variety of HCC cells using RT-qPCR, and consistent with the bioinformatics results, PCAT6 was indeed aberrantly highly expressed in HCC. Because of the highest expression of PCAT6 in HepG2 cells, we selected the HepG2 cells with the highest expression to design two siRNAs and used RT-qPCR to determine the effective siRNA for the functional assay. In HepG2 with siRNA knockdown of PCAT6, the proliferation and migration abilities of HepG2 cells were significantly reduced. All these results suggest that PCAT6 may be a prognostic marker and therapeutic target associated with copper death in HCC.

There are some deficiencies and limitations in this study. On the one hand, we only used the data in the TCGA database. Although there are 424 samples in HCC, the limited number of samples leads us to miss some potential biomarkers. In addition, our study lacked validation of the second cohort. Although we searched the GEO and ICGC databases, we did not find a suitable dataset to validate our model due to imperfect sequencing platforms and lack of clinical information, and we will explore it further in a subsequent study. On the other hand, the mechanism of cuproptosis is complicated, and we only found that PCAT6 may interact with cuproptosis-related genes, such as LIPT1, LIPT2, SLC31A1, GLS, etc.; however, there is a long way to go to investigate which molecules PCAT6 interacts with, and this requires more in-depth and extensive experiments to explore. We have also verified the function of PCAT6 in HCC by RT-qPCR and functional assays at the cellular level; however, their precise molecular regulation mechanism still needs to be corroborated by further studies, and more clinical samples are required to confirm it.

In summary, our study provides a reliable method for prognosis prediction in HCC patients and helps to elucidate the network and mechanism between cuproptosis-related genes and lncRNAs. Furthermore, the predictive model could sensitively identify HCC patients who responded better to immunotherapy. Among seven lncRNAs, PCAT6 may be a prognostic marker and therapeutic target associated with copper death in HCC.

## 4. Materials and Methods

### 4.1. Preparation of Transcriptomic, Mutation Data, and Clinical Information

The transcriptomic data and clinical features of LIHC patients were downloaded from TCGA (https://portal.gdc.cancer.gov/ (accessed on 6 January 2023)), including 374 tumor samples and 50 normal samples. Strawberry Perl (version 5.30.0.1-64bit, https://strawberryperl.com/, (accessed on 6 January 2023)) and R were used to extract and distinguish the transcripts per kilobase million and the complete pathological information of each clinical sample. We downloaded the clinical information of 377 patients from TCGA, of which only 376 patients had a survival time and survival status. The transcriptional data of 424 samples were downloaded from TCGA, including 50 normal samples and 374 tumor samples. The clinical information of 376 patients was combined with 424 samples of TCGA transcription data, and 370 sample data were finally collected after removing the normal samples and the duplicate data.

### 4.2. Screening of Cuproptosis-Related lncRNAs

Nineteen cuproptosis-related genes were collected from the literature, and co-expressed lncRNAs associated with cuproptosis were obtained by Pearson correlation analysis. The parameters were set to |pearson| > 0.3 and *p*_value_ < 0.001 [47,48,49,50,51].

### 4.3. Establishment of the Risk Model Based on Cuproptosis-Related lncRNAs

We calculated univariate Cox regression analysis using survival packages in R to screen for prognostic cuproptosis-related lncRNAs (*p*_value_ < 0.05). Then, the lasso regression was conducted by the glmnet package in R (the penalty parameter was determined by 10-fold cross-validation). During lasso regression, the genes with the risk of overfitting were removed according to the partial likelihood bias and lambda value, lncRNAs were obtained with a significant difference with OS, and the tuning parameter was set to lambda.min. We used multivariate Cox regression by the survival packages in R to screen lncRNA again to determine the lncRNA for the final construction model. In detail, the split ratio between the training and testing sets was investigated via the analysis of the models with three different ratios (5:5, 7:3, 8:2). Appendix A indicated that the risk prognostic model with the ratio of 5:5 exhibited better accuracy, stability and comprehensiveness than that with the ratios of 7:3 and 8:2, thus the ratio of 5:5 was used in this study. The samples from TCGA were randomly divided into a training set (185 samples) and a testing set (185 samples). We established three risk prediction models for the total sample, training group, and test group. The training set was used to construct the prognostic model for cuproptosis-related lncRNAs, and the testing set was used to verify and optimize the prognostic model. The risk score can be calculated by the following equation [52]:(1)$$risk\;scores=\sum_{i=1}^{n}coef\;\left(lncRNAi\right)\times expr\;\left(lncRNAi\right)$$
where *coef* and *expr* denote the coefficients and the expression of lncRNAs, and *coef* is *β* in the multivariate Cox regression analysis formula. In the multivariate Cox regression analysis model, *β* is the regression coefficient. For genes with a high *β* value, the danger (risk of death) is higher, so the prognosis is worse.
(2)$$h\left(t\right)=h_{0}\left(t\right)e^{\beta 1x1+\ldots \ldots +\beta nxn}$$

On this basis, all the patients were divided into high-risk and low-risk groups according to the median risk score.

### 4.4. Validation of the Risk Model

PCA and risk scores correlation analysis were conducted to verify the ability of the risk model to distinguish between high-risk and low-risk groups. The C-index and ROC curve by the survival, survminer, and timeROC packages in R were used to estimate the accuracy of the risk model. The clinical characteristics (age, gender, stage, T stage, M stage, and N stage) were considered in univariate and multivariate Cox regression analyses by the survival package in R to determine whether the risk score was independent of clinical characteristics. The prognosis prediction of the risk model was analyzed by Kaplan–Meier survival analysis by the survival and survminer package in R.

### 4.5. Function Analysis

GSEA is a gene-set-based enrichment analysis method [30]. Thus, the differences in signaling pathways between high-risk and low-risk groups were analyzed using GSEA. We downloaded the “c2. cp. kegg. v7.1. symbols” gene set from the MSigDB database which is available on the GSEA website (https://www.gsea-msigdb.org/gsea/downloads.jsp (accessed on 6 January 2023)). The NES is the corrected ES value that is normalized by the data from the gene set. Pathways with NES > 1 and *p*_value_ < 0.05 may be statistically significant signal pathways.

### 4.6. Immune Response and Analysis

The immune infiltration results based on TIMER, CIBERSORT, quantTIseq, xCell, MCP-counter, and EPIC algorithms [53,54,55,56,57,58] of the tumor were downloaded from TIMER2.0 (http://timer.comp-genomics.org/, (accessed on 6 January 2023)). Estimation of stromal and immune cells in malignant tumours using expression data (ESTIMATE) enables the estimation of stromal and immune cells in malignant tumor tissues [59,60]. We further used the ESTIMATE algorithm to calculate the ImmuneScore, StromalScore, and EstimateScore of the risk model population. We used the Maftool package in R 4.1.3 software to analyze and visualize genes from the top 15 tumors’ mutational burden in both high- and low-risk groups. A single sample gene set enrichment analysis (ssGSEA) algorithm was used to calculate immune checkpoint levels in different grouped samples. ssGSEA is a GSEA analysis for a single sample. The R package used is GSVA, and the gsva function specifies method = “ssgsea”. Additionally, TISIDB [61] (http://cis.hku.hk/TISIDB/download.php, (accessed on 6 January 2023)) was used to analyze the immune infiltration, and TIDE were applied to demonstrate the immune response.

### 4.7. Exploration of Potential Antitumor Drugs

To predict the potential drugs for the treatment of patients with HCC, the R package pRRophetic was used to predict the drug response to the expression matrix, with *p*_value_ < 0.001 as the screening criterion.

### 4.8. Quantitative Real-Time PCR (RT-qPCR)

Total RNA was isolated using the Trizol kit (Takara, Shiga, Japan) and reverse transcribed using the Takara RT reagent (Takara, Shiga, Japan). Quantitative real-time PCR was also performed using Takara RT reagents (Takara, Shiga, Japan). The upstream and downstream primer sequences for PCAT6 were: 5′-CCCCTCCTTACTCTTGGACAAC-3′ (forward) and 5′-GACCGAATGAGGATGGAGACAC-3′ (reverse). For GAPDH: 5′-ACCCAGAAGACTGTGGATGG-3′ (forward) and 5′-TTCAGCTCAGGGATGACCTT-3′ (reverse). All results were processed by the 2^−ΔΔCT^ method after normalization to GAPDH. All the experiments were repeated three times.

### 4.9. Cell Culture and Transfection

Normal human hepatocytes (LO2) and two HCC cell lines (HepG2 and Hep3B) were representative in cancer studies and were obtained from the Cell Bank of Shanghai Institutes of Biological Sciences, Chinese Academy of Sciences. The cells were cultured in DMEM with 10% fetal bovine serum (Gibco, Rockville, MD, USA), 1% penicillin-streptomycin solution (Gibco, San Diego, CA, USA) in a humidified incubator at 37 °C, 5% CO_2_, except for Hep3B, which was cultured in MEM medium with 10% fetal bovine serum (Gibco, CA, USA), 1% penicillin-streptomycin solution (Gibco, CA, USA) in a humidified incubator at 37 °C, 5% CO_2_. The medium was renewed every two days.

Transient silencing of small interfering RNA (siRNA) from PCAT6 was achieved using a siRNA duplex. SiRNA was purchased from GENEWIZ (Suzhou, China). The siRNA was transfected into the indicated cells using Lipofectamine 2000 (Invitrogen, Waltham, MA, USA), followed by RT-qPCR to verify the efficacy of the interference [62].

### 4.10. Cell Counting Kit-8 Experiment

Cell Counting Kit-8 (Yeasen Biotechnology, Shanghai, China) was used to measure the effect of siRNA knockdown of PCAT6 on the cell proliferation capacity. HepG2 cells transiently transfected with siRNA for PCAT6 (siPCAT6) or control siRNA (si-NC) were inoculated into 96-well plates at 2 × 10^3^. Absorption values were measured 24, 48, 72, and 96 h after siRNA transfection, according to the kit instructions. Data are shown as mean ± SEM. All the experiments were repeated three times.

### 4.11. Wound Healing

HepG2 cells transiently transfected with siRNA for PCAT6 (siPCAT6) or control siRNA (si-NC) were inoculated into six-well plates and then incubated at 37 °C and 5% CO_2_ until they reached 90% fusion. A 200 µL pipette tip was then used to produce a straight-line scratch in the cell monolayer. After washing away the cell debris with PBS, the cells were incubated under normal conditions. Images of the scratched area (wound) were taken under a microscope at time points 0, 24, and 48 h, respectively. Data are shown as the relative distance between the two edges. Scratch healing rate = (original wound area-existing wound area)/original wound area. The wound healing area was calculated by ImageJ2. All the experiments were repeated three times.

### 4.12. CFSE Assay

Huh7 cells was digested with trypsin and washed twice with PBS buffer, then resuspended Huh7 cells with PBS. CFSE was added into cells with the final concentration of 5 μM, incubating for 20 min at 37 °C. After 20 min, 40% serum was added into the cells, incubating for 20 min at 37 °C. Cells was washed twice with complete medium (DMEM + 10% FBS + 1% PS). Cells were inoculated into twelve-well plates at the density of 3 × 10^4^ per well, and different concentrations of Sorafenib (0 μM, 2.5 μM, 5 μM, 10 μM) were added after 12 h. After 72 h, cells were collected and wash twice with PBS, performing flow assay.

### 4.13. Cell Cycle Assay

Huh7 cells were inoculated into six-well plates at the density of 1 × 10^5^ per well, and different concentrations of sorafenib (0 μM, 2.5 μM, 5 μM, 10 μM) were added after 12 h. Cells were collected after 12 h and were resuspended with 75% ethanol and fixed at −20 °C for 24 h. After 24 h, cells were washed twice with PBS, PI staining solution (SIMUBIOTECH, Tianjin, China) was added, and the cells were stained for 30 min at room temperature for flow-through PE channel assay.

### 4.14. Statistical Analysis

Statistical analysis was performed using Graph Pad Prism 9. T-tests were used to assess differences between any two groups of data. All data are expressed as mean ± SEM. A *p*_value_ < 0.05 was considered a significant difference.

## 5. Conclusions

In this study, a risk model associated with cuproptosis was constructed in HCC based on seven lncRNAs. The Kaplan–Meier, PCA, ROC, C-index, and nomogram results suggested that this risk model maybe an independent prognostic predictor. The high-risk group also responded more sensitively to immunotherapy. PCAT6 may be a prognostic marker and therapeutic target associated with cuproptosis in HCC; however, further experimental verification is needed for more in-depth studies.

## Figures and Tables

**Figure 1 ijms-24-09987-f001:**
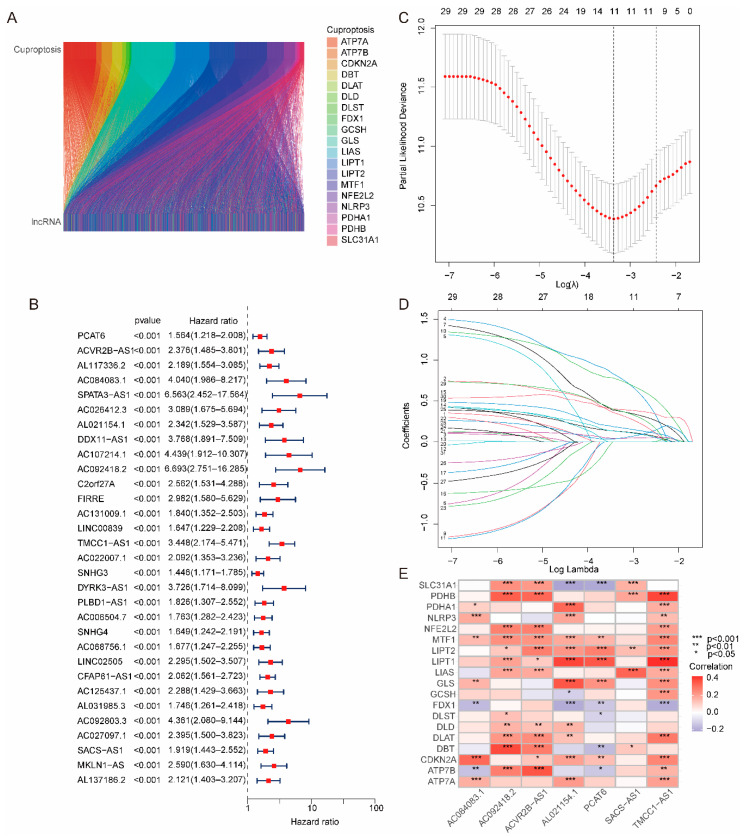
Construction of risk model based on cuproptosis-related genes. (**A**) Sankey diagram of cuproptosis-related genes and lncRNAs. (**B**) A total of 31 lncRNAs were determined to be associated with prognosis by univariate Cox regression analysis. (**C**) Error curve of cross-validation. (**D**) Coefficient profiles were drawn based on (log *λ*) sequences and the value of lambda.min was defined based on 10-fold cross-validation, where the optimal *λ* yielded 11 lncRNAs. (**E**) Heatmap of the association between 19 cuproptosis-related genes and 7 lncRNAs.

**Figure 2 ijms-24-09987-f002:**
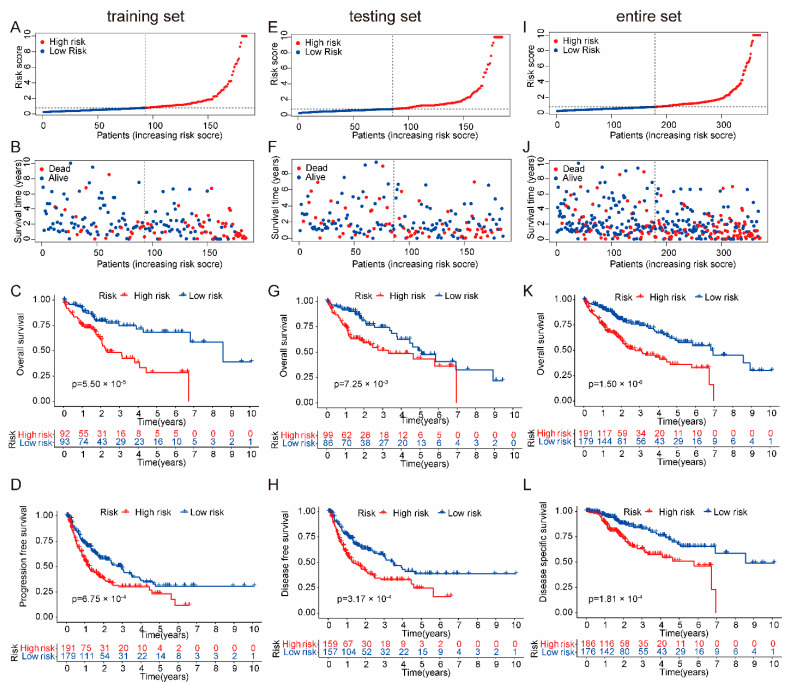
Grouping and prognostic of the risk model. (**A**) The ranked scatter plots in the TCGA training set. (**B**) The survival state diagram plots in the TCGA training set. (**C**) The Kaplan–Meier survival curves plots in the TCGA training set. (**D**) The ranked scatter plots in the TCGA testing set. (**E**) The survival state diagram plots in the TCGA testing set. (**F**) The Kaplan–Meier survival curves plots in the TCGA testing set. (**G**) The ranked scatter plots in the TCGA entire set. (**H**) The survival state diagram plots in the TCGA entire set. (**I**) The Kaplan–Meier survival curves plots in the TCGA entire set. (**J**) The Kaplan–Meier survival curves plots. The PFS of patients in the high-risk group had significantly lower survival rates than those in the low-risk group. (**K**) The Kaplan–Meier survival curves plots. The DFS of patients in the high-risk group had significantly lower survival rates than those in the low-risk group. (**L**) The Kaplan–Meier survival curves plots. The DSS of patients in the high-risk group had significantly lower survival rates than those in the low-risk group.

**Figure 3 ijms-24-09987-f003:**
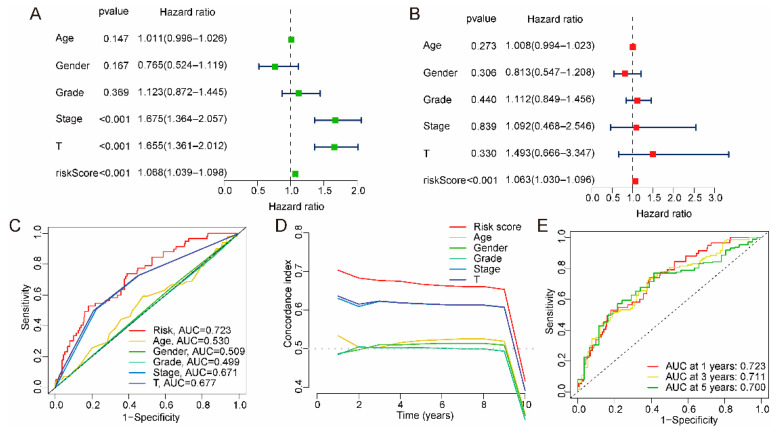
Validation and estimation of the risk model independent of other clinical characteristics. (**A**,**B**) The risk model prognostic factors could be independent prognostic factors through univariate and multivariate Cox regression analyses. (**C**) ROC curve plots. The accuracy of risk models outperforms other clinical characteristics. (**D**) C-index. The accuracy of risk models outperforms other clinical characteristics. (**E**) ROC curve plots. The accuracy of risk models for patients at 1, 3, and 5 years.

**Figure 4 ijms-24-09987-f004:**
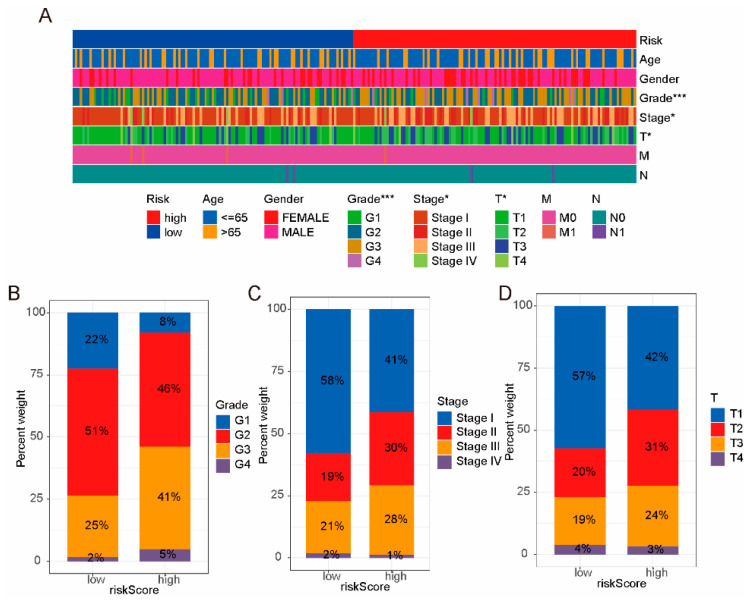
The correlation between risk model and clinical characteristics. (**A**) Heatmap of risk model and clinical characteristics. The risk model was associated with grade, stage, and T stage. (**B**–**D**) The high-risk group had higher grade, stage, and T stage of HCC. * *p*_value_ < 0.05, *** *p*_value_ < 0.001.

**Figure 5 ijms-24-09987-f005:**
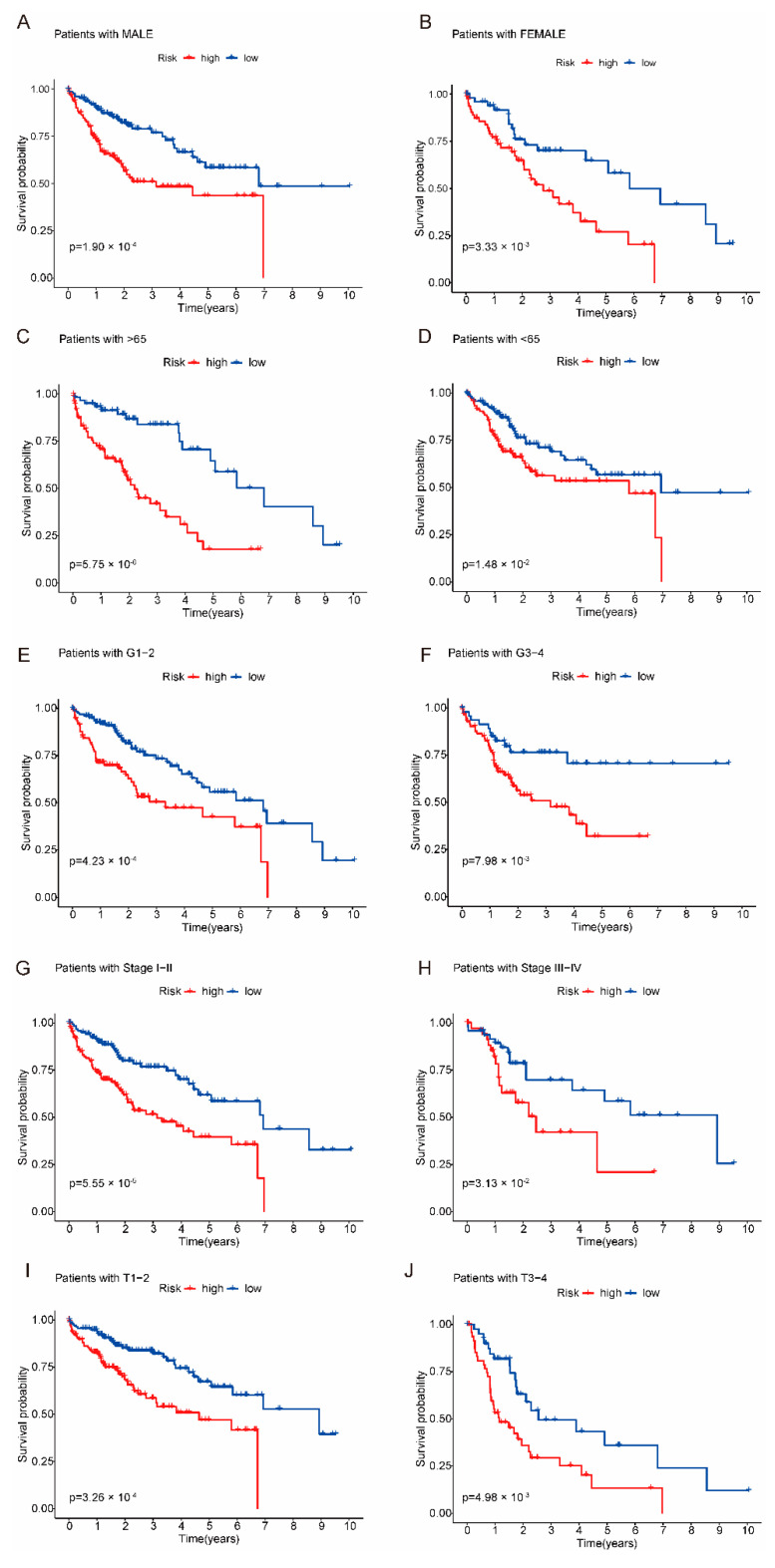
Kaplan–Meier survival curves of clinical characteristics. The survival rate of the high-risk group was significantly lower than that of the low-risk group. (**A**) The Kaplan–Meier survival curves plots. The male of patients in the high-risk group had significantly lower survival rates than those in the low-risk group. (**B**) The Kaplan–Meier survival curves plots. The female of patients in the high-risk group had significantly lower survival rates than those in the low-risk group. (**C**) Patients with aged > 65 years had significantly lower survival rates than those in the low-risk group. (**D**) Patients with aged < 65 years had significantly lower survival rates than those in the low-risk group. (**E**) Patients with grade 1–2 had significantly lower survival rates than those in the low-risk group. (**F**) Patients with grade 3–4 had significantly lower survival rates than those in the low-risk group. (**G**) Patients with stage I–II had significantly lower survival rates than those in the low-risk group. (**H**) Patients with stage III–IV had significantly lower survival rates than those in the low-risk group. (**I**) Patients with T stage 1–2 had significantly lower survival rates than those in the low-risk group. (**J**) Patients with T stage 3–4 had significantly lower survival rates than those in the low-risk group.

**Figure 6 ijms-24-09987-f006:**
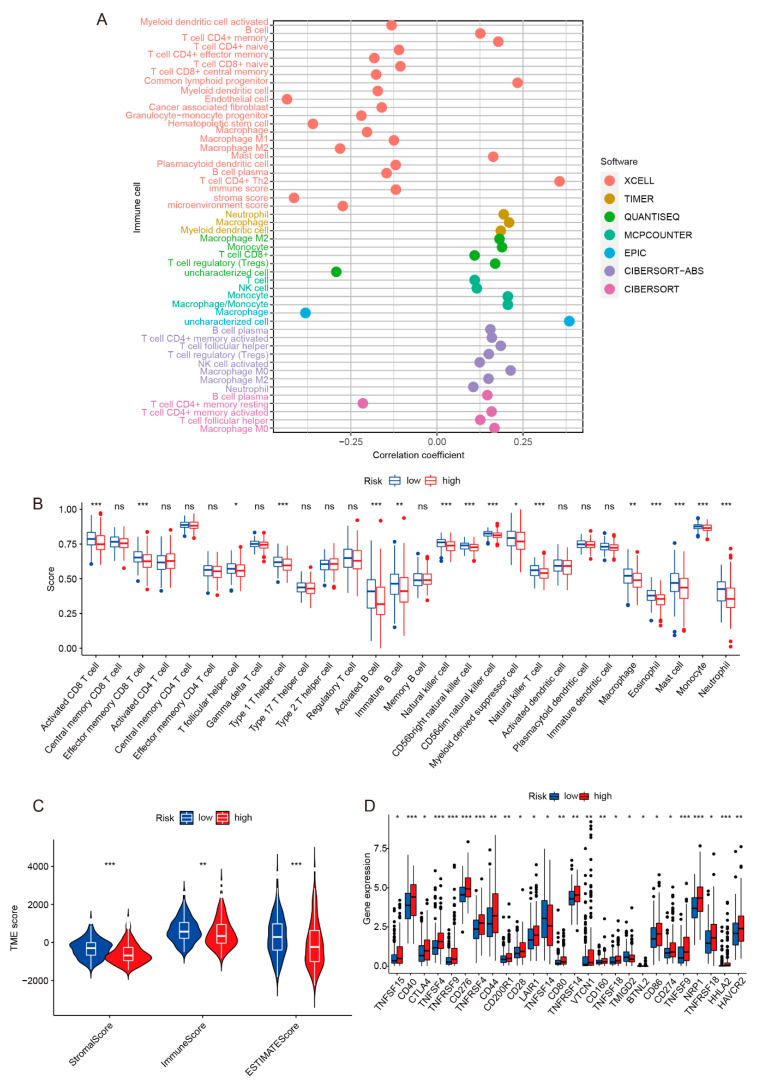
Assessing the immune infiltration of the risk model. (**A**) The immune infiltration correlation result was based on TIMER, CIBERSORT, quantTIseq, xCell, MCP-counter, and EPIC algorithms. (**B**) ssGSEA algorithm was applied to analyze tumor infiltrating lymphocytes (TILs). (**C**) The tumor microenvironment (TME) was evaluated by using the ESTIMATE algorithm. The high-risk group had significantly lower immune scores than the low-risk group. (**D**) The difference analysis of immune checkpoint genes between high-risk and low-risk groups. * *p*_value_ < 0.05, ** *p*_value_ < 0.01, *** *p*_value_ < 0.001.

**Figure 7 ijms-24-09987-f007:**
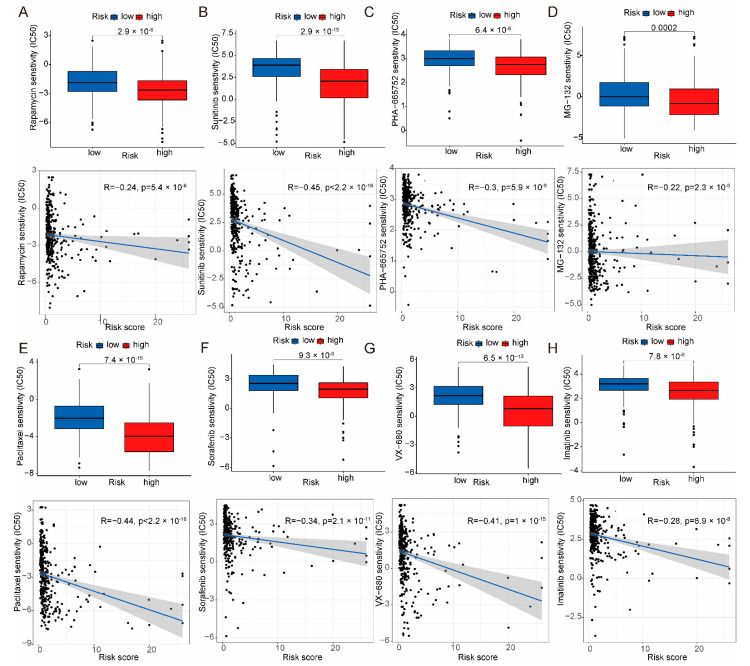
The pRRophetic algorithm was used to predict drug sensitivity in the risk model. (**A**–**H**) The boxplot described the sensitivity of IC_50_ in the risk model, and the correlation graph revealed the relationship between the risk model score and IC_50_.

**Figure 8 ijms-24-09987-f008:**
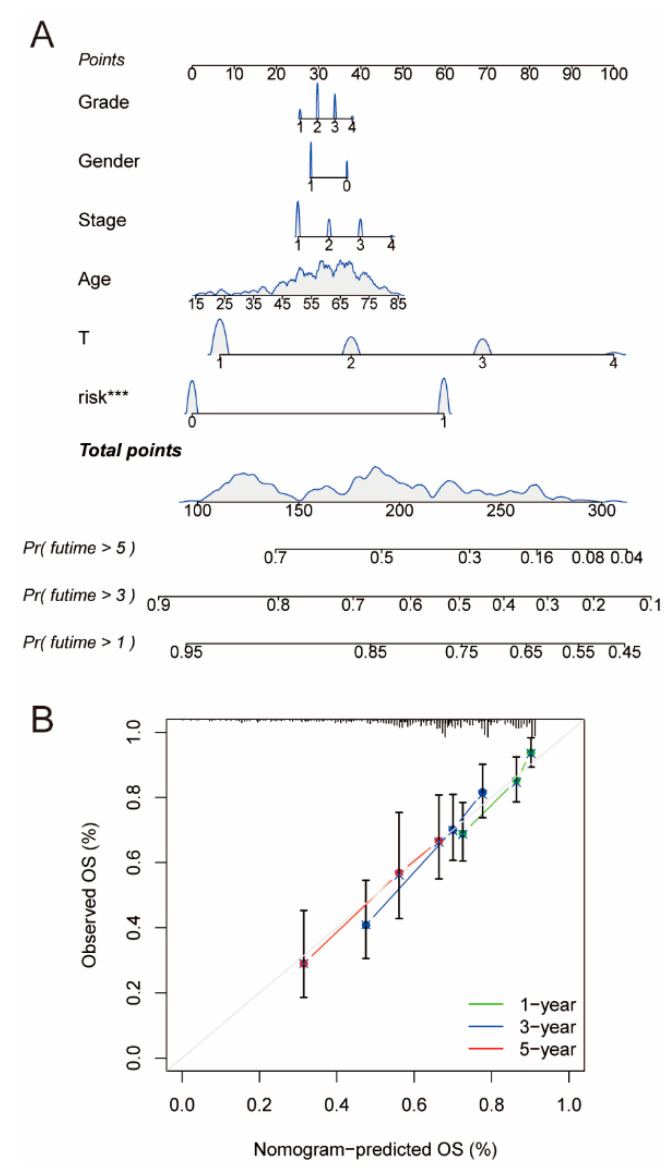
Construction and evaluation of prognostic nomogram. (**A**) The prognostic nomogram combined with the risk model and clinical characteristics. A nomograph is a diagram that is based on multivariate regression analysis, integrates more than one prediction indicator, and then makes use of scaled line segments to draw on the same plane at a sure scale to categorize the interrelationships between more than a few variables in the prediction model. The names of the nomograms normally encompass three categories: the top points (scores), the variable names in the middle prediction model (grade, gender, stage, age, T stage, risk), and the prediction probability of survival for 1, 3, and 5 years. (**B**) The calibration plot of the nomogram. *** *p*_value_ < 0.001.

**Figure 9 ijms-24-09987-f009:**
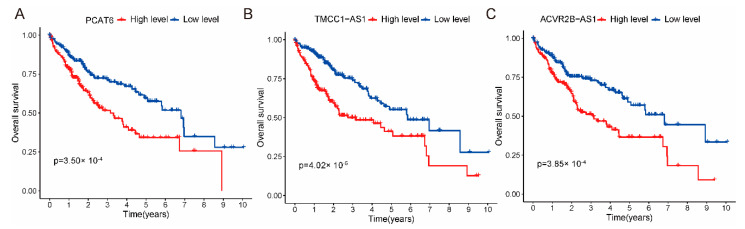
The Kaplan–Meier survival curves. (**A**–**C**) The expression level of lncRNAs (TMCC1-AS1, PCAT6, ACVR2B-AS1) was significantly associated with the prognosis of HCC patients (*p*_value_ < 0.001).

**Table 1 ijms-24-09987-t001:** Differences in clinical characteristics between training and testing sets.

Covariates	Type	Total	Testing	Training	*p* _value_
Age	≤65	232 (62.7%)	118 (63.78%)	114 (61.62%)	0.7471
Age	>65	138 (37.3%)	67 (36.22%)	71 (38.38%)	
Gender	FEMALE	121 (32.7%)	59 (31.89%)	62 (33.51%)	0.8246
Gender	MALE	249 (67.3%)	126 (68.11%)	123 (66.49%)	
Grade	G1	55 (14.86%)	26 (14.05%)	29 (15.68%)	0.6769
Grade	G2	177 (47.84%)	87 (47.03%)	90 (48.65%)	
Grade	G3	121 (32.7%)	66 (35.68%)	55 (29.73%)	
Grade	G4	12 (3.24%)	5 (2.7%)	7 (3.78%)	
Grade	Unknown	5 (1.35%)	1 (0.54%)	4 (2.16%)	
Stage	Stage I	171 (46.22%)	85 (45.95%)	86 (46.49%)	0.9642
Stage	Stage II	85 (22.97%)	43 (23.24%)	42 (22.7%)	
Stage	Stage III	85 (22.97%)	44 (23.78%)	41 (22.16%)	
Stage	Stage IV	5 (1.35%)	3 (1.62%)	2 (1.08%)	
Stage	Unknown	24 (6.49%)	10 (5.41%)	14 (7.57%)	
T	T1	181 (48.92%)	92 (49.73%)	89 (48.11%)	0.5286
T	T2	93 (25.14%)	44 (23.78%)	49 (26.49%)	
T	T3	80 (21.62%)	40 (21.62%)	40 (21.62%)	
T	T4	13 (3.51%)	9 (4.86%)	4 (2.16%)	
T	Unknown	3 (0.81%)	0 (0%)	3 (1.62%)	
M	M0	266 (71.89%)	139 (75.14%)	127 (68.65%)	1
M	M1	4 (1.08%)	2 (1.08%)	2 (1.08%)	
M	Unknown	100 (27.03%)	44 (23.78%)	56 (30.27%)	
N	N0	252 (68.11%)	136 (73.51%)	116 (62.7%)	0.7399
N	N1	4 (1.08%)	3 (1.62%)	1 (0.54%)	
N	unknown	114 (30.81%)	46 (24.86%)	68 (36.76%)	

## Data Availability

Not applicable.

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
