# Peer review of "Creation of a Prognostic Model Using Cuproptosis-Associated Long Noncoding RNAs in Hepatocellular Carcinoma"

_ijms, 2023, doi:10.3390/ijms24129987_

Round 1
Reviewer 1 Report
The submitted manuscript entitled “Construction of LIHC prognostic model based on the cuproptosis-related lncRNA and PCAT6 maybe a biomarker verified by bioassay” focuses
on constructing a risk model associated with cuproptosis in hepatocellular carcinoma. Authors also identified PCAT as a potential biomarker for predicting prognosis in LIHC This study scientifically sounds and can be of interest for the journal audience. The manuscript contains (55 relevant references total) and more than 50% of relevant references published last 5 years and 13 Figures and two Tables to illustrate the results obtained. However, there are some concerns and recommendations to improve the quality of the manuscript. There are as follows:
1. The title of the manuscript should be corrected. The usage of the next abbreviation (LIHC) looks incorrect. HCC is more appropriate.
2. The Abstract section is large. It must be reduced.
3. On the contrary, the Introduction section should be expanded.
4. There is almost no information about PCAT6 and/or other possible biomarkers in the Introduction section. .
5. There are mistakes in the text. For example: “PACT6 may be a prognostic marker and therapeutic target associated with cuproptosis in LIHC” in the Conclusions section.
Author Response
Dear Reviewer:
Thank you very much for your valuable suggestions for our work. We have carefully considered and made revisions, and you can see the specific response content in the attachment.
With best regards,
Sincerely Yours
Jianping Lin

Reviewer 2 Report
Dear authors,
Your paper has too many things inside which are not related to each other. I was expecting a paper on cuproptosis and found everything else except this. First of all, what you are presenting is more about copper metabolism in liver cancer, not cuproptosis. Keep it simple. Explore the TCGA and then validate your findings. You cannot put together so much in silico data and then inhibit a gene and look only at a proliferation and migration assay. This is not a validation at all. The wet lab experiments have nothing to do with the aim of your work. I haven't seen a single experiment showing a cuproptosis mechanism, not even a cell death experiment. Define clearly your ideas, the aim, test and validate it, also with lab experiments and then try and publish..
I have made a few comments in the pdf, but unfortunately the amount of work required to order this manuscript and transform it in a publishable research article is far beyond the possibility of any peer review process. A lot of work has to be done and I wish you the best of luck! I am sure that you will be able to get there!

Author Response

(The authors gave the same response as above.)

Reviewer 3 Report
Please refer to the attached file.

Author Response

(The authors gave the same response as above.)

Round 2
Reviewer 1 Report
I am sure that the authors have made all to improve the manuscript quality to be published in IJMS. The manuscript may be published in IJMS in the present form.
Author Response
Dear reviewer,
Thanks a lot for your valuable comments and suggestions, this paper has greatly improved, thank you very much!
With best regards,
Sincerely Yours
Jianping Lin
Reviewer 2 Report
Dear Authors,
Cuproptosis is a mechanism of cell death. How is that in line with what you say?: high concentrations of copper ions can promote tumor proliferation, invasion, and angiogenesis, leading to cell cuproptosis, which is relevant to cancer development.
Please find the comments in the attached pdf.
Other considerations:
The title should be changed. maybe something like this might be of your liking: Creation of a prognostic model using cuproptosis associated Long Non-Coding RNAs in Hepatocellular Carcinoma
Do not use present time. Patients HAD a worse prognosis. not HAVE.
The conclusions drawn are a bit too strong given the lack of validation of the results on a second cohort, and the lack of in vitro exploration of the mechanisms associated with the findings.
I suggest at least adding some experiments to explore the predictive value of the drug sensitivity prediction.
Best of luck!

Reviewer 3 Report
Thanks for revising the manuscript.
Author Response

(The authors gave the same response as above.)
